# A Narrative Review of Occupational Air Pollution and Respiratory Health in Farmworkers

**DOI:** 10.3390/ijerph18084097

**Published:** 2021-04-13

**Authors:** Kayan Clarke, Andres Manrique, Tara Sabo-Attwood, Eric S. Coker

**Affiliations:** Environmental and Global Health Department, University of Florida, Gainesville, FL 32603, USA; clarkek@ufl.edu (K.C.); mepx56@phhp.ufl.edu (A.M.); sabo@phhp.ufl.edu (T.S.-A.)

**Keywords:** air pollution, farmworkers, respiratory health, vulnerable group, environmental epidemiology, occupational exposure

## Abstract

The agricultural crop sector in the United States depends on migrant, seasonal, and immigrant farmworkers. As an ethnic minority group in the U.S. with little access to health care and a high level of poverty, farmworkers face a combination of adverse living and workplace conditions, such as exposure to high levels of air pollution, that can place them at a higher risk for adverse health outcomes including respiratory infections. This narrative review summarizes peer-reviewed original epidemiology research articles (2000–2020) focused on respirable dust exposures in the workplace and respiratory illnesses among farmworkers. We found studies (*n* = 12) that assessed both air pollution and respiratory illnesses in farmworkers. Results showed that various air pollutants and respiratory illnesses have been assessed using appropriate methods (e.g., personal filter samplers and spirometry) and a consistent pattern of increased respiratory illness in relation to agricultural dust exposure. There were several gaps in the literature; most notably, no study coupled occupational air exposure and respiratory infection among migrant, seasonal and immigrant farmworkers in the United States. This review provides an important update to the literature regarding recent epidemiological findings on the links between occupational air pollution exposures and respiratory health among vulnerable farmworker populations.

## 1. Introduction

The agricultural industry plays a fundamental role in determining the vitality of a nation’s economy and its accessibility to nutritious food [1]. In the United States, agricultural productivity increased six-fold in the 20th century through the mechanization of agriculture, concentrated animal feeding operations (CAFOs), and the use of agrochemicals in crop farming [2]. Moreover, as the demand for agricultural production has increased, the U.S. agriculture industry has become progressively dependent on large-scale farms, with family-owned small-scale farms making up only 21% of all agricultural production [3]. These dramatic shifts from small-scale family-owned crop farms to large-scale crop farms have led to a hired workforce that is highly dependent on migrant, seasonal and immigrant farmworkers. Despite these major changes in agricultural production in the U.S., the legal framework regulating the workplace continues to be based on the family farm labor model, which entails limited laws regulating worker safety, compensation, and employee benefits in agriculture [2,3].

The U.S. agricultural industry employs approximately 5 million workers, acting as one of the major contributors to the nation’s economy [4,5,6]. Approximately 2.4 million individuals employed in the agricultural crop workforce identify themselves as young Hispanic migrant, seasonal, and immigrant farmworkers (MSIFWs) [7]. The 2007–2009 National Agricultural Workers Survey (NAWS) noted that 48% of farmworkers lacked legal documentation—making it very difficult to obtain accurate farmworker population estimates in the U.S. [4]. Despite this uncertainty regarding the true size of the agricultural workforce, the composition of the U.S. agricultural sector is known to rely heavily on what is largely an immigrant workforce [2]. It is universally recognized that MSIFWs are one of the most vulnerable populations globally and in the U.S.; the MSIFW population faces a myriad of social injustices, health disparities and social inequalities [4,6]. They have limited access to quality healthcare, and experience financial insecurity and high rates of poverty. MSIFWs also face language and other cultural barriers, crowded living conditions, and a general lack of legal support and political influence [4,7,8]. These issues are compounded by the under-reporting of workplace hazards and injuries that is related to fear of deportation tied to their illegal immigration status or fear of potential loss of work based on their precarious employment [8]. Moreover, the significant reduction in the domestic agricultural workforce over the past few decades has paralleled the funding available for agricultural health research leading to large gaps in occupational epidemiological research with MSIFWs communities [9]. These gaps limit our current understanding of the state of occupational hazards among farmworkers, such as occupational air pollution exposure and respiratory health risks. Additionally, the shift to a largely MSIFWs population in the U.S. has raised additional barriers to acquiring reliable health surveillance data because of their near-constant mobility, lack of healthcare, and evasion of federal authorities.

Mortality surveillance data among individuals, 25 years and older, with asthma and chronic obstructive pulmonary disease (COPD) in the U.S. show a higher proportionate mortality ratio among certain industries including agriculture [10]. Previous studies demonstrate that farmworkers are exposed to a combination of physical, chemical, biological, and psychosocial stressors that can influence their vulnerability and susceptibility to disease [5,8]. Susceptibility in this review refers to compromised innate or adaptive immunity that enhances the virulence of a pathogen [11]. In general, farmworkers are exposed to more elevated levels of particulate air pollution compared to their general population counterparts [12,13,14,15]. An emerging body of the literature suggests that exposure to particulate matter (PM) sized 2.5 micrometers or less increases the risk of ARI and all-cause mortality in the general population [16], yet occupational exposure of crop farmworkers to PM_2.5_ and the association with ARI risk is not well studied.

The last extensive review on the respiratory health of farmworkers that we identified was published in 1998 by Schenker et al. [2]. This report noted large gaps in respiratory health epidemiologic research relevant to agricultural worker health which included limited application of occupational exposure assessment methods for respirable mineral dust, aerosolized chemicals, and monitoring for respiratory pathogens. While Schenker describes studies that show farmworker susceptibility to infections from pathogens that are related to agricultural practices such as contact with animals (e.g., zoonoses), the literature is lacking on farmworkers’ susceptibility to respiratory infections from pathogens that are transmitted person to person in the general population, such as influenza.

The present study provides a narrative review of the peer-reviewed literature since the extensive review was published by Schenker 20 years ago. Based on the previously identified gaps and our current knowledge of farmworker respiratory health, we set out to address the following questions: (1) How have researchers assessed respirable dust exposure among farmworkers? (2) What methods have been used to assess respiratory infections and what types of respiratory pathogens have been assessed among farmworker populations? (3) Are there epidemiological studies that have investigated the association between respirable dust exposures and respiratory infections in farmworkers?

## 2. Materials and Methods

We conducted a systematic review of the peer-reviewed literature. Our approach included only English-language journal articles. We searched for original research articles in the PubMed and Web of Science databases using the following Boolean search terms: “respiratory illness”, “respiratory virus”, “particulate matter”, “air pollution”, “agricultural dust”, “influenza” and “farmworkers” or “agricultural workers”. Initial screening of studies for inclusion was based on reading abstracts from the papers that were retrieved from the database searches and determining (a) if farmworkers were participants in the study and (b) if the study assessed any type of respiratory health outcome. We used the U.S. National Institute of Occupational Safety and Health definition of “farmworker” as any agricultural worker who works on either a livestock farm or a crop farm [17]. Since part of our review is intended to address exposure assessment for studies that investigated the relationship between occupational air pollution exposure and respiratory health among farmworkers, we conceptualized respiratory health broadly in our review. Specifically, we identified respiratory health studies reporting health status that relates to any part (upper or lower) of the respiratory tract such as a diagnosis of a chronic respiratory health condition (e.g., asthma or COPD), acute or chronic respiratory infections, lung function measurements, etc. [18]. We further evaluated articles meeting these initial screening criteria to determine if occupational air pollution exposure was assessed in the study. We define occupational air pollution exposure assessment broadly as any type of questionnaire-based or instrument-based qualitative (e.g., high versus low) or quantitative (air concentration) measure of air pollution exposure in the agricultural workplace environment. If we established that an assessment of occupational air pollution exposure was conducted in the study, we then determined if that study calculated a measure of association between air pollution exposure and a respiratory health outcome. Such studies constituted original occupational epidemiology studies that investigated the relationship between air pollution exposure and respiratory health in farmworkers. We note that we did find a number of research journal articles on farmworkers that assessed knowledge attitudes and practices (KAPs) of farmworkers, review articles of farmworker respiratory health, and review articles of farmworker exposure to air pollution. We excluded these articles because either they were not occupational epidemiology studies, as defined above, or they were not original research articles. Some studies were either vaccination or pesticide exposure specific studies that had no mention of respiratory health, which were also excluded from our review. 

Figure 1 shows that a total of 203 studies were retrieved from the publication databases, of which 157 were from PubMed and 46 from Web of Science. After merging and de-duplicating of articles, 152 unique studies remained. After applying the inclusion and exclusion criteria described above, a total of 12 studies were included and reviewed completely.

As outlined in the introduction, the purpose of this review is to qualitatively describe occupational air pollution exposure assessment methods used for respiratory health studies among farmworkers, and to identify how respiratory infections have been assessed and to understand if the relationship between occupational air pollution exposures and respiratory infections has been studied among crop farmworkers. Hence, the focus of our systematic review is narrative and not a meta-analysis. Here, we describe the study populations, study designs, statistical methods, types of exposure measurement used in the study, results, and strengths and limitations of the included studies. A major motivation behind our review is to identify gaps in research data regarding susceptibility to respiratory infections among crop farmworkers due to occupational air pollution exposures, as well as possible gaps in occupational air pollution exposure assessment for respiratory health epidemiology studies conducted among crop farmworkers. Therefore, we have summarized the prevalence of studies that investigate respiratory infections, studies that performed personal air monitoring in the workplace, and studies that focused on crop farmworkers versus livestock farmworkers.

## 3. Results

Of the twelve papers under review, nine (75%) represented cross-sectional studies [19,20,21,22,23,24,25,26,27], and three (25%) were prospective studies [28,29,30]. The publication years spanned from 2004 to 2018 among the studies, 67% (*n* = 8) of which were published between 2013 and 2018.

As indicated in Figure 2, the most common occupation of the study population was crop farmworkers (46%, *n* = 5) [20,21,22,27,30], followed by livestock farmworkers (31%, *n* = 4) [19,25,26,29], then both livestock and crop farmworkers (15% *n* = 2) [23,28] and one study that did not specify the type of farmworker occupation [24]. Half of the included studies were conducted in the United States [22,23,24,25,26,30] and the other half of the studies were conducted among farmworkers in European countries including France [28,29], Poland [27], Portugal [19] and Denmark [21], and other areas such as the Middle East [20,21] and Southeast Asia [21]. Of all the studies conducted in the U.S., just two indicated that they focused their study among immigrant farmworkers [26,30].

### 3.1. Assessment of Occupational Exposure to Air Pollution

The studies included in this analysis used a variety of methods to assess air pollution, including personal and stationary air monitoring instruments, questionnaires and semi-structured interviews, as well as surface sampling of inhalable dust. Table 1 summarizes the sampling equipment used, frequency of measurement, what was being measured and the range of PM concentration for each study if it was quantified. Six studies [19,21,23,25,26,27] used personal air monitors with filters to assess air pollution exposure in the occupational setting. One study [29] coupled personal air monitoring and questionnaires to assess occupational exposure to air pollution. The questionnaires collected data about the farm’s characteristics such as the building and cleaning practices, and these data were used as predictive variables in mixed effects models [29]. Two studies used stationary filter air monitors only [20,28]. One study set up air monitors in the center of four agricultural villages; the villages were spread over a 65 km^2^ area [20]. Another study conducted filter air sampling in the participants’ homes and workspaces, at a height that was away from any obvious source of pollution [28]. One coupled stationary sampling, at the four corners of the greenhouse and one in the center, with surface dust sampling conducted using sterile cotton swabs to assess air pollution in the workplace [22]. Two studies used questionnaires only to assess air pollution exposure [24,30]. Researchers asked questions regarding dust exposure and agricultural work history and calculated a time-weighted self-reported average (TWSRA) dust score for exposed workers [30] in one of the studies and the other calculated dust exposure qualitatively ranging from “no”, “low” to “high”, based on time in dusty environments and hours per week performing farm work [24].

Studies have measured a number of air pollutants, from particulate matter to volatile organic compounds, endotoxins, fungi, bacteria and inhalable microorganisms. Five studies quantitatively measured particulate matter (PM) [19,20,23,26,28], sized 0.5 micrometers (μm), 1 μm, 2.5 μm, 5 μm and 10 μm, as well as inhalable PM which was defined as sized 100 μm or less [26]. Five studies measured endotoxins [22,23,25,26,27], which is a component of gram-negative bacterial cell walls. One study measured muramic acid, which is a component of gram-positive bacteria cell walls [25]. One study measured volatile organic compounds such as hexane, benzene, ethylbenzene, trichloroethylene, toluene, tetrachloroethylene, decane isomers, butoxyethyl acetate and undecane isomers [28]. Three studies extracted fungi and bacteria (gram-negative and -positive) from filter air samples [21,22,27]. Two studies [21,22] measured β-glucan and β-d-glucan, which are fungal antigens found in common fungal infections. Three studies [21,22,27] also measured inhalable airborne microorganisms. Three studies [24,29,30] measured respirable dust in general.

### 3.2. Respiratory Health Outcome Assessment

Figure 3 and Figure 4 summarize the types of methods used to assess respiratory health and the types of conditions assessed to determine respiratory health status. The most common method used to assess respiratory health in the 12 studies was self-report using questionnaires (*n* = 10) [19,20,22,23,24,25,26,27,28,29] and spirometry measurements of lung function (*n* = 6) [19,20,25,26,28,30] (Figure 4). Another method used to assess respiratory health was nasal lavage (*n* = 2) [21,23]. Nasal lavage is also known as a nasal wash, where a saline solution is squeezed into the nose and immediately collected for pathogen detection. One study conducted physician medical check-ups (*n* = 1) [28], which involved clinical measurements such as blood pressure, pulse, saturation, lung function testing, and prick testing for common allergens, as well as bio-specimen samples including blood, urine, and saliva.

Self-report of respiratory symptoms, such as cough, wheezing, and shortness of breath, were the most commonly assessed respiratory conditions (26%) (Figure 4). Other commonly reported respiratory conditions included pulmonary function (23%) determined from spirometry measurements (Figure 4). Other respiratory conditions assessed included self-reported diagnosis for asthma (*n* = 6) [19,20,22,24,28,29], chronic bronchitis (*n* = 4) [22,24,28,29], chronic obstructive pulmonary disease (COPD) (*n* = 2) [20,28], emphysema (*n* = 2) [22,23], pneumonia (*n* = 1) [22], and allergen prick test (*n* = 2) [27,29]. A more detailed breakdown of respiratory health illnesses is provided in Figure 4. 

### 3.3. Summary of Epidemiological Findings

Epidemiologic data from these studies showed similarities concerning the association between occupational exposure to air pollution and respiratory health outcomes. These findings are grouped by farmworker type to allow for ease of reference and comparison among respiratory outcome in similar farmworker groups. Table 2 summarizes the study design and key epidemiologic findings from each reviewed study.

#### 3.3.1. General (Unspecified) Farmworkers

A cross-sectional study was conducted among California primary farm operators (PFO) to characterize respiratory health of farmworkers and their occupational risks for respiratory disease [24]. The population studied were majority white (84.5%) males (89.9%) with a median age of 54 years old [24]. Logistic regression analysis showed that a higher concentration of dust was associated with higher persistent wheeze by 1.8 (1.1–3.2) fold and chronic bronchitis by 1.8 (0.9–3.7) [24].

#### 3.3.2. Crop Farmworkers

A cross-sectional study found that higher endotoxin exposure was associated with a two- to three-fold increase in mean nasal lavage fluid polymorphonuclear neutrophils (PMN), myeloperoxidase (MPO), albumin, and eosinophilic cation protein (ECP) levels among grain handlers, cattle feedlot workers, dairy workers and corn farmers across Colorado and Nebraska compared to other farmers in the same study with lower endotoxin exposure [23]. In a study conducted in California among Mexican immigrant vegetable and fruit farmers, there was a significant association between longer time in agriculture and worse lung function (FVC and FEV1/FEV6) [30]. The same study also noted a significant relationship between elevated dust score and lung function (FEV6) [30]. Among greenhouse (plants and flowers) workers in the Midwestern USA, the prevalence of self-reported respiratory symptoms (asthma, wheezing, phlegm, hay fever, cough) was higher in the greenhouse workers compared to a control group of office workers [22].

A study conducted among farmworkers who worked in stables and granaries in France used multiple regression models to show statistically significant negative association between exposure to occupational PM_2.5_ and reduced cytokine levels [28]. Among greenhouse workers from Central and Northern Europe, the Middle East, and Southeast Asia, researchers found a significant correlation between exposure to fungi and β-glucan and its content in nasal lavage for the male participants [21]. There was not a significant association noted for unspecified bacteria [21]. Logistic regression showed that increased PM_2.5_ levels increased the risk for burning in the mouth, nose, and throat by 2.3-fold, increased the risk of burning symptoms in the eyes by 2.6-fold, increased wheezing by 2.1-fold, and increased chest tightness by 2.2-fold [21].

A study among cotton farmworkers in Turkey also showed that increased occupational PM_10_ exposure levels were associated with higher chest tightness, and PM_2.5_ exposure was significantly associated with wheezing and chest tightness using multivariable linear regression analysis [20]. A study conducted in Poland among hop growers observed that lung function was significantly lower in farmworkers compared to the control group of office workers adjusting for gender, height and smoking (469.7 +/− 127.5 vs. 562.9 +/− 123.8; *p* < 0.001) [27].

#### 3.3.3. Livestock Farmworkers

Among dairy farmworkers in California, the majority being Hispanic/Latino males, with mean age 33 years, it was found that lung function was lower with a higher concentration of inhalable particles and total endotoxin exposure (not significant) [26]. Among a similar population of dairy farmworkers across four states located in North Central U.S., there was a notable association between higher occupational exposure to inhalable endotoxin and lower lung function [25]. A study conducted in Brittany, France found that cough and phlegm were significantly associated with long-term exposure to occupational respirable dust among egg production workers [29]. The same study also found chronic bronchitis was 4-fold higher when exposure levels of respirable dust were greater than 0.1 mg/m^3^ [29].

A study conducted in Portugal among poultry farmworkers observed that the prevalence of obstructive pulmonary ventilatory disturbances was greater in farmworkers with longer occupational exposure and saw a notable but non-significant trend for higher concentration of PM and higher self-reported frequency of upper and lower respiratory symptoms [19].

## 4. Discussion

Occupational health surveillance data (1988–1998) suggest that, relative to agricultural workers from other sectors (e.g., horticulture, forestry), agricultural crop farmworkers have a higher risk of mortality from acute respiratory infections (ARIs) for both upper and lower respiratory infections [31]. In the ongoing COVID-19 pandemic, there have been several case clusters among crop farmworkers infected with the novel coronavirus (SARS-CoV-2) that have greatly impacted migrant communities within just days of the first cases diagnosed in the U.S. [32,33]. This helps reveal that there are likely underlying risk factors that influence the transmission, infectivity and severity of respiratory pathogens in the crop farmworker populations that have yet to be studied. In this review, we aimed to understand the ways in which occupational epidemiologic studies have assessed respirable dust exposure, how respiratory infections have been assessed, and if there were studies that looked at the relationship between PM_2.5_ and ARIs among farmworkers. Overall, our review of the literature shows that no studies have investigated the link between occupational exposure of PM_2.5_ and ARIs among farmworkers. Given the existing evidence on the possible causal role of PM_2.5_ on ARI severity as well as surveillance data suggesting increased ARI severity among crop farmworkers, it is imperative that we begin to research the effects of occupational particulate matter exposure on susceptibility to severe acute respiratory infections among crop farmworkers. 

### 4.1. Summary of Results and General Findings

The twelve studies identified in this narrative systematic review included farmworkers from the crop and livestock sectors. These 12 studies were conducted predominantly in North America and Europe. Methods employed to assess respiratory outcomes mostly involved biologic measurements (spirometry and nasal lavage) or questionnaires to assess the prevalence of specific respiratory conditions (e.g., asthma diagnosis). While personal air sampling methods were predominantly employed in these studies, we did not find any studies that characterized PM chemical composition. The 12 studies consistently showed that farmworkers occupationally exposed to elevated levels of respirable dust had higher prevalence of respiratory symptoms and adverse conditions such as chronic bronchitis and asthma, and decreased lung function.

A lack of studies that characterize PM chemical composition is an important gap identified in our review of the literature. In general, respirable dust in the agricultural environment is a heterogeneous mixture of organic and inorganic airborne particles that are generated from various farming practices [5,6,8,34,35,36]. While several included studies employed filter-based sampling to detect various PM size fractions or the presence of endotoxins in PM, these approaches lack the chemical compositional information needed to understand the true nature of occupational health risks posed by respirable dust exposure among farmworkers. In addition to respirable dust, farmworkers can be exposed to an array of other air contaminants such as toxic gases, bioaerosols, and endotoxins that act as respiratory irritants, leaving them at a higher prevalence of both acute and chronic respiratory disease [2]. The development of more protective occupational health regulations for vulnerable farmworker populations would greatly benefit from more focused epidemiologic data that characterize the chemical composition of PM exposures among the agricultural workforce. Characterization of PM composition could focus on contaminants such as crystalline silica, metals, and chemical pesticides.

The cross-sectional study design predominated in the included studies. We assume this to be the case because cross-sectional studies are cheaper and more time efficient compared to prospective cohort designs [37]. The results from cross-sectional studies are less generalizable and it is difficult to infer the temporal association between occupational exposure and respiratory health outcomes [37]. However, results from the reviewed studies that employed a cross-sectional design can inform hypotheses and assumptions for prospective (or retrospective) cohort studies. We emphasize that in the U.S. crop farmworker context, retrospective cohort studies may not be feasible due to a general lack of historical employment records among migrant, immigrant and seasonal farmworkers (MISFWs). We therefore recommend prospective cohort studies that longitudinally examine ARI incidence and repeat measures of occupational air pollution exposure. 

### 4.2. How Have Researchers Assessed Respirable Dust Exposure among Farmworkers?

Of the studies that employed air monitors, all used filter-based air monitors to characterize the type of air pollution the farmworkers were exposed to in the workplace. More than half (58%) of the studies used personal air monitors, which is supported by the OSHA recommendation regarding the accuracy of air pollution exposure in the breathing zone of workers in an occupational setting [38].

Even though stationary air monitors are known for their accuracy and precision in characterizing air pollutants, their fixed location limits the ability to infer occupational exposures [39]. While personal filter-based sampling is considered the “gold standard” in occupational air pollution exposure assessment, there are newer, more discrete sampling technologies that could be worth exploring for occupational exposure assessment among MISFWs. One such technology is the Ultrasonic Personal Air Sampler (UPAS), which is a filter-based personal sampler that is more light-weight, smaller, and quieter compared to conventional cyclone samplers used for filter sampling [40]. Additionally, the UPAS (Access Sensor Technologies, Fort Collins, CO, USA) does not require the use of an external pump or tubing and provides real-time measurements of temperature and humidity conditions and GPS data logging capabilities. We have recently deployed UPAS samplers among immigrant greenhouse workers and migrant strawberry picking farmworkers in Florida. The UPAS was well received in the study population. Additionally, discrete, real-time nephelometer samplers that are coupled with a smartphone app [41] can be useful for empowering MISFWs to participate in air pollution exposure monitoring studies without the fear of scrutiny from their employers or fellow workers. Overall, respirable dust was measured among farmworkers using appropriate devices and methods. However, research is recommended in terms of evaluating newer devices that may be more widely accepted among crop farmworkers who face precarious working conditions. 

Apart from the types of samplers used for occupational air pollution exposure assessment, it is also important to discuss which pollutants have been assessed. The air pollutants measured in the included studies include particulate matter of various size fractions (0.5 micrometers to 100 micrometers), fungi, bacteria (Gram-negative and Gram-positive), and endotoxins and its 3-hydroxy fatty acid (3-OHFA). Of the air pollutants assessed in the included studies, pollutant levels were in general at higher concentrations compared to ambient air pollutant levels. There are other occupational airborne pollutants relevant for study in this population, such as fibers from certain crops that are small enough to evade an individual’s non-specific immune respiratory response, from crops such as cotton, hemp, flax, coconut husks, and pineapple [42,43].

### 4.3. What Methods Have Been Used to Assess Respiratory Infections and What Types of Respiratory Pathogens Have Been Assessed among Farmworker Populations?

Respiratory infection is caused by viral and bacterial agents in the upper or lower respiratory tract [44]. Based on this definition, we have established that none of the studies measured bacterial or viral respiratory infection. Instead, the studies focused on acute and chronic respiratory symptoms that are indicative of a compromised respiratory state, while a few studies focused on self-reported diagnosis of a chronic respiratory illness. Acute respiratory symptoms included wheeze, phlegm, cough and shortness of breath, which are among the common symptoms of acute respiratory infections [45]. The chronic respiratory disease outcomes measured in the selected studies include asthma, chronic bronchitis, chronic obstructive pulmonary disease (COPD), and lung cancer, among others. All of the assessed symptoms or conditions were either self-reported or diagnosed by a medical professional; the latter is considered more accurate yet less commonly applied among the studies selected in this literature review. A major limitation of self-reporting of respiratory symptoms is information bias [46]. Information bias in epidemiological studies can lead to inaccurate estimates of association, or over- or underestimation of risk parameters; however, when performed correctly, self-reporting is more useful in giving a range of responses than other data collection instruments [46]. Self-reporting, therefore, would be more appropriate for open-ended questions in qualitative research designs, which was not the case for any of the studies selected in this literature review.

Other studies went beyond questionnaire-based self-reported experience of respiratory symptoms and employed spirometry testing, which is a lung function test used to assess how well a person’s lungs work by measuring how much air they inhale, how much and how quickly they exhale [47]. It is also one of the most commonly used methods to diagnose respiratory conditions such as asthma and chronic bronchitis [48]. However, this approach is challenging because the results of spirometry testing are dependent on proper calibration of the instrument, proper training of personnel who are conducting the test, and selecting an appropriate reference population [47]. Despite these challenges, current practice dictates that spirometry testing is the most appropriate tool to differentiate between normal lung function and obstructive and restrictive respiratory diseases; however, it requires proper handling of the data and use of the spirometer to provide accurate results.

### 4.4. Are There Epidemiological Studies That Have Investigated the Association between Respirable Dust Exposures and Respiratory Infections in Farmworkers?

We did not find any studies that addressed this question of occupational exposure to respirable dust and respiratory infection risk among farmworkers. Nor did any of the included studies assess any respiratory infections, neither viral nor bacterial, as part of their research into the associations between respirable dust and respiratory health. There are existing studies that investigated bacterial and viral respiratory infections among farmworkers. These studies, however, are seroprevalence studies of swine [49,50] or avian influenza [51,52,53], tuberculosis [54,55], or other bacterial infections such as leptospirosis [56,57] and *Escherichia coli* [58]. We emphasize that despite some research into respiratory infections among farmworkers, none of the studies have explicitly investigated an association between these infections and occupational exposure to respirable dust, such as PM_2.5_.

When inhaled, PM_2.5_ can deposit deep into the lung where it may evade the body’s nonspecific human defenses and subsequently, cause injury directly to the lungs or can impact other organ systems such as the cardiovascular system. However, the precise role and associated mechanisms by which PM_2.5_ enhances ARI susceptibility are unknown, but several studies suggest particles may impair innate and adaptive immunity of the respiratory system [59,60,61,62]. For example, work by our group has shown in vitro and in vivo that exposure to nanoparticulates can lead to enhanced influenza A viral titers in concert with reduced innate immune pathways that specifically control host defenses (e.g., Pattern recognition receptor pathways) [63,64,65]. In another study, long-term PM_2.5_ inhalation lowers the capacity of pulmonary macrophages to secrete IL-6 and IFN-β, a disorder in the pulmonary innate defense system which results in increased death rates following influenza infection [66]. The composition of the PM can be highly variable and can include various metals and particle-bound polycyclic aromatic hydrocarbons, which have the capacity to increase the production of free radicals, consume antioxidants and cause oxidative stress [67]. Using inhibitors of oxidative stress, our group has determined a partial role for oxidative stress produced by carbon nanoparticulates in the increased influenza viral titers observed in lung cells [64]. The presence of PM_2.5_ in the lower respiratory system can also cause an imbalanced intracellular calcium homeostasis and inflammatory injury [67]. These effects on the respiratory system leaves an individual’s immune response compromised against possible infection by virus or bacteria. Another example of how the exposome impacts disease severity among farmworkers is occupational exposure to endotoxin found in respirable dusts. When inhaled, endotoxin is deposited in the airways and is taken up by a lipopolysaccharide binding protein and destroyed by a macrophage [68]. Once the macrophage internalizes the endotoxin, a variety of inflammatory cytokines are produced including IL-1ß, TNF-a and IL-6 [68]. Infection of viral or bacterial pathogen on already inflamed airways can have an additive effect or exacerbate lung pathology [69]. It is also important to consider the impact of multiple, cumulative exposure to stressors on the severity of health outcomes. The exposome, defined as the total exposure that humans experience from the moment of conception, birth and throughout adulthood, may be particularly critical for farmworkers [70]. In addition to enhanced susceptibility from agricultural PM_2.5_ exposure, farmworkers can experience specific vulnerabilities that could exacerbate ARI severity. A lack of access to health care in rural areas among crop farmworkers can make this population vulnerable to less access to timely diagnostic tests. Language barriers experienced by MISFWs can also influence the quality of care once they do gain access to health care. Cost barriers to health care and medications can also arise among farmworkers because of high levels of poverty and a lack of health insurance coverage. Crowded housing conditions in migrant farmworker camps can also influence pathogen exposure and transmission risk. Hence, the question remains as to whether a combination of chemical and non-chemical environmental stressors could be playing a significant role in the observed ARI mortality risks among U.S.-hired crop farmworkers. Despite these concerns, we identified only two studies that focused on the immigrant farmworker population, which highlights the lack of research for this vulnerable population. The studies that did focus on the MISFW population were both in California [71]. However, there are many hired immigrant farmworkers across the U.S., such as the Midwest and South, who have not been studied to understand the effects of this type of occupational exposure on respiratory health [71]. Our review of the literature reveals a critical need for these questions to be addressed in future occupational epidemiological studies among crop farmworkers.

### 4.5. Other Gaps Identified

There is a noticeable lack of studies from developing countries in South America, Africa and Asia, which limits the generalizability of the current literature in a global health context [19,21,25]. While there are other studies conducted in the farmworker population from these lower income regions [72,73], they tend to focus on other matters such as seroprevalence of types of influenza or risks attributable to climate change [74]. More studies need to be conducted in areas outside of North America and Europe.

There is also the need for more specific assessment of respiratory health outcomes, either through serology or biomarkers. The use of spirometry is appropriate but is open to user and evaluation errors; however, with properly trained personnel and equipment calibration, these errors can be avoided. Finally, there is a need for more epidemiological studies to assess respirable dust as part of the exposome and its effects on respiratory infections by capitalizing on current technology to identify suitable biomarkers for exposure and health risk.

### 4.6. Limitations and Strengths

This narrative review has limitations and strengths. Firstly, journals outside of PubMed and the Web of Science database could have been missed in our review. Even though this is an important limitation, we believe the themes and gaps identified will remain constant with added studies. Another limitation is that we did not conduct a meta-analysis which would have quantitatively determine effect size across studies. Even though our motivation here is to simply provide a narrative review to highlight the current state of the literature, there is still a need for a meta-analysis to quantitatively assess effect sizes of occupational air pollution exposures on respiratory health outcomes among farmworkers. The strength of our review is that it synthesizes a broad range of occupational epidemiology studies on the relationship between respirable dust and respiratory illnesses in the farmworker population, which enables us to highlight important gaps and provide recommendations of opportunities to conduct further investigations in this area of occupational exposure among this vulnerable population.

## 5. Conclusions

The public health issue of respiratory infections and occupational air pollution exposure is important and the need to foster work on this topic is great, especially for vulnerable crop farmworker populations. In particular, there is a need for studies that couple assessment of respirable dust and respiratory infections, whether bacterial or viral, to determine the level of association between these two factors among crop farmworkers. We show that the agricultural occupational health literature is consistent with respect to positive relationships between occupational dust exposure and respiratory symptoms, which suggests a strong link to higher respiratory infection severity risks.

## Figures and Tables

**Figure 1 ijerph-18-04097-f001:**
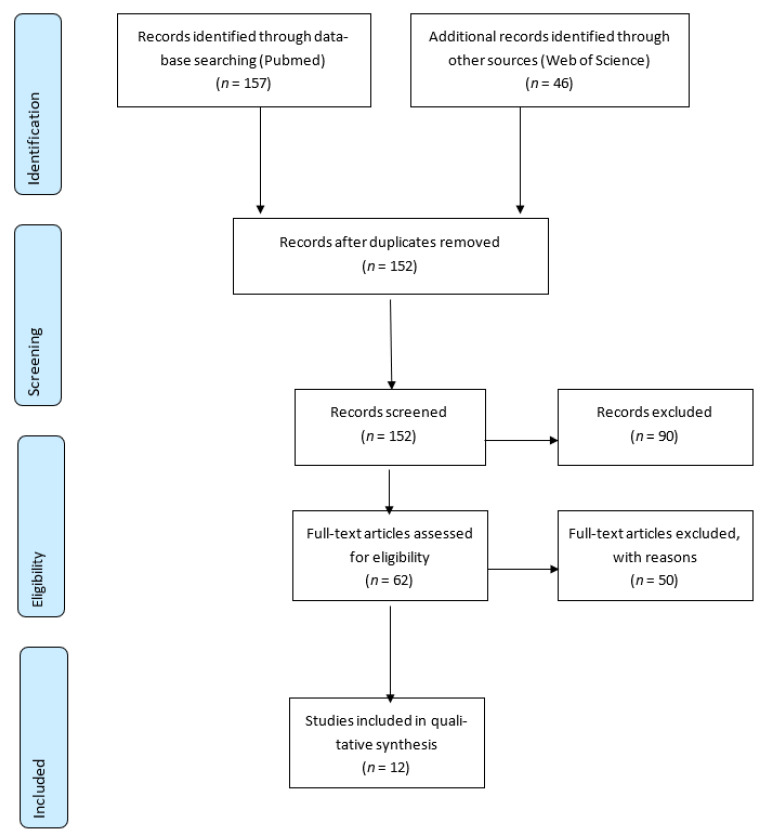
Flow diagram of inclusion and exclusion criteria at each phase of the data mining process leading to the final work retained.

**Figure 2 ijerph-18-04097-f002:**
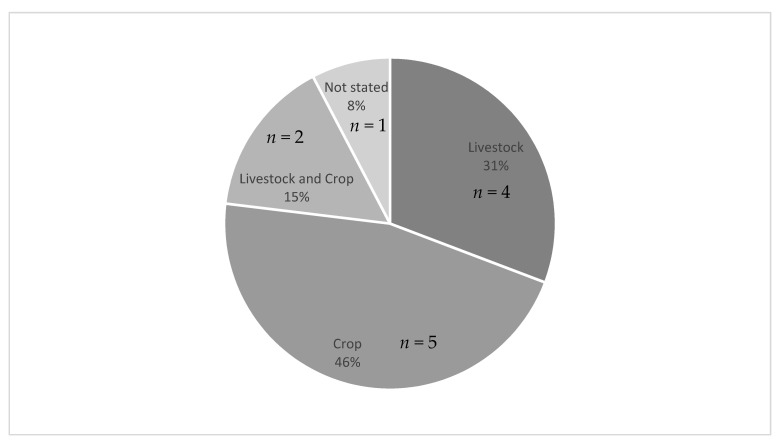
Type of farmworker occupation for included studies (*n* = 12).

**Figure 3 ijerph-18-04097-f003:**
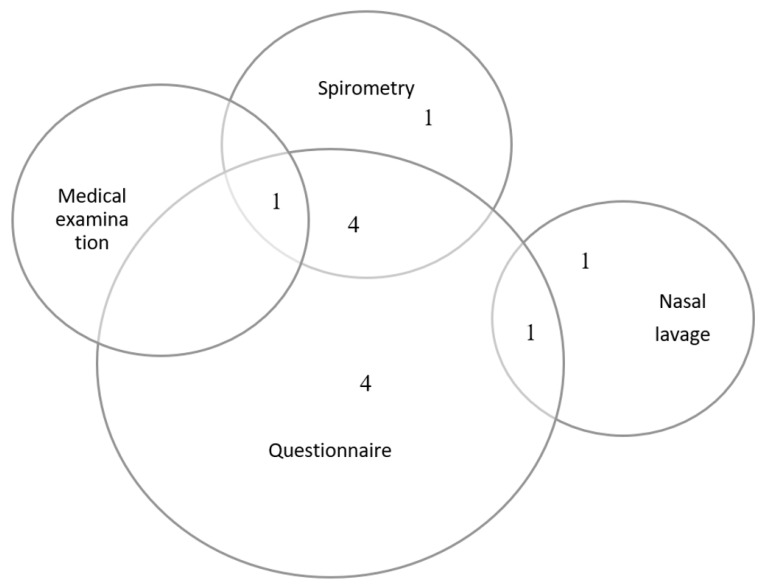
Respiratory health assessment methods.

**Figure 4 ijerph-18-04097-f004:**
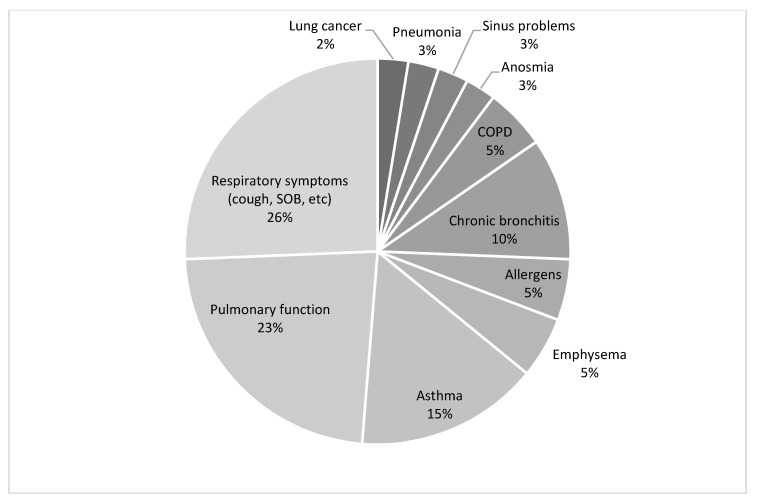
Respiratory condition assessed.

**Table 1 ijerph-18-04097-t001:** Exposure assessment measurement tools, frequency and chemical composition of variable of interest.

Study	Type of Farmworker and/or Crop (N)	Type of Air Sample	Specific Instrument	Work Shift Exposure Measurement	Frequency of Measurement	Contaminant Type	PM2.5 Concentration Range
Mitchell et al. (2015) [26]	Dairy farmers (N = 205)	Personal filter sample	SKC button sampler for collecting inhalable PM (<100 μm in aerodynamic diameter) onto a Teflon 25-mm Millipore PTFE filter, with a pore size of 3.0 μm (Fisher FSLW02500). A GK2.05SH (KTL) cyclone sampler (BGI Inc., Waltham, MA) collected particles with a cut point of 2.5 μm (PM2.5) onto a Teflon filter (Fisher, FHLP03700).	Yes	Not stated	PM	Geometric mean concentration PM = 812 μg/m^3^ PM2.5 = 35 μg/m^3^
Nonnenman et al. (2017) [25]	Dairy parlor workers (N = 62) A parlor is a building where cows are milked on a dairy farm.	Personal filter sample	Inhalable dust (50% cut-point at 100 μm) was sampled in the worker’s breathing zone using an inhalable sampler (Button Aerosol Sampler, SKC Inc. manufacturer, Eighty Four, Pennsylvania, USA) and personal sampling pumps (AirChek XR 5000, SKC Inc.)	Yes	Duration of a single work-shift	Inhalable dust (<Xµm) Muramic acid Endotoxin	NA
Burch et al. (2009) [23]	Grain elevator workers, cattle feedlot, dairy and corn farm workers (N = 125)	Personal filter sample	Personal breathing zone samples for inhalable particulate matter were collected using (IOM) sampling cassettes and 25mm PVC filters with a 5-μm pore size (SKC, Eighty Four, PA).	Yes	Duration of a single work-shift	Inhalable dust Endotoxin Endotoxin’s 3-hydroxy fatty acid (3-OHFA)	NA
Guillam et al. (2013) [29]	Egg production workers (N = 100)	Personal filter sampler	Personal dust sampler, 11 ozs, 10 lmp flowrate—CIP 10—ARELCO	Yes	Cold season work-shift Warm season work-shift	Respirable dust (<4 µm in diameter)	NA
Góra et al. (2004) [27]	Crop farmers (hop growers) (N = 69)	Personal filter sample	AP-2A personal sampler—TWOMET, Zgierz, Poland at the flow rate of 2 L/min. Glass fiber filters, with 1 µm pore size and 37 mm diameter.	Yes	Once during harvest season	Airborne microorganisms, dust and endotoxin Gram-positive bacteria (corynebacteria and bacilli) Fungi (Penicillium citrinum, Alternaria alternata, and Cladosporium epiphyllum) Thermophilic actinomycetes Gram-negative bacteria	NA
Madsen et al. (2013) [21]	Greenhouse vegetable crop workers (N = 33)	Personal filter sample	Gesamtstaubprobenahme (GSP) inhalable samplers—Gesamtstaubprobenahme by BGI, Inc., Waltham, MA, USA; polycarbonate filter (pore size 1 µm)	Yes	Sampling took place from 6:00 or 7:00 to 15:00 or 16:00 during the Wednesdays immediately preceding the Thursday of nasal lavage sampling	Fungi Bacteria β-glucan	NA
Viegas et al. (2013) [19]	Poultry farm workers (N = 47)	Personal filter sample	Portable direct reading equipment—Lighthouse, model 3016 IAQ	Yes	During performance of different tasks in pavilions	PM0.5 PM1 PM2.5 PM5 PM10	PM0.5 = 2.8–25 µg/m^3^ PM1 = 4.7–32 µg/m^3^ PM2.5 = 20–240 µg/m^3^ PM5 = 220–2400 µg/m^3^ PM10 = 1400–15,200 µg/m^3^
Adhikari et al. (2011) [22]	Greenhouse (flowers and plants) workers (N = 49)	Stationary monitoring	Button Inhalable Aerosol Samplers—SKC, Inc., Eighty Four, PA, USA	Yes	During winter and summer for 5 to 7 h per one work shift—four from the corners of the greenhouses and one from the center.	Fungi Bacteria Actinomycetes Endotoxin (1→3)-β-D-glucan	NA
Audi et al. (2017) [28]	Granary and stable workers (N = 72)	Stationary monitoring	Radiello Passive Sampler (for BTEX) to measure VOCs Aerocet 530 device to measure PM2.5	Yes	Assessed over a 3-month period with samplers placed where participants are expected to spend the greatest number of hours, as well inside granaries and stables.	VOCs include hexane, benzene, ethylbenzene, trichloroethylene, toluene, tetrachloroethylene, decane isomers, butoxyethyl acetate and undecane isomers. Fine particles (PM2.5)	Highest mean value of PM2.5 (11 µg/m^3^), with the highest median value (3 µg/m^3^) and the highest third quartile value (8 µg/m^3^)
Sak et al. (2018) [20]	Persons living in cotton-farming villages (N = 252)	Stationary real-time with gravimetric validation	pDR 1500 Thermo Scientific Personal Data RAM pDR device and two cyclones were used to make PM10 and PM2.5 measurements	Yes	Before and after pesticide application. Fifteen-minute measurements of PM10 and PM2.5 were made with the cyclones. Measurements were made at the four village centers (in four villages spread over 65 km^2^) before agricultural spraying (in mid-June) and within 15 min and 48 h after agricultural spraying (in mid-August).	PM10 PM2.5	PM10 = 11.7–334.8 µg/m^3^ PM2.5 = 4.7–17.2 µg/m^3^
Schenker et al. (2005) [24]	Primary farm operator (PFO) (N = 100)	Questionnaire	Farmers were asked the following question: “In the past year, approximately what percentage of the time that you spent farming did you spend working at a dusty job?” They were also asked to report the numbers of hours they personally worked on their farm operation over the last year (by season) and the percentage of time spent in the general categories of administrative, field, and livestock tasks. Dust exposure variables were considered “none”, “low” or “high” based on percent time in dust multiplied by average yearly hours per week farming, and percent time in dusty environment.	NA	NA	Dust	NA
Rodriguez et al. (2014) [30]	Mexican migrant crop (melons, tomatoes, nuts, grapes, cotton, lettuce, asparagus, onion, pomegranate, etc.) farmers (N = 450)	Questionnaire	Time-weighted self-reported average (TWSRA) dust scores were calculated for dust exposure in a year by multiplying the number of weeks a participant worked in each crop type and job task combination by the average number of days worked per week. Next, the number of days worked for each crop type and job task combination was multiplied by its corresponding self-rated dust score.	NA	NA	Dust score	NA

**Table 2 ijerph-18-04097-t002:** Key findings of the included studies.

Study	Study Design	Demographic Characteristics of Study Population	Statistical Analysis	Key Findings—Odds Ratio (95% CI) or β (95%CI)
Góra et al. (2004) [27]	Cross-sectional study	53.6% male Median age 48 years Non-smokers 42%	Spearman test	Positive correlation between exposure to airborne endotoxin and IL-6 level in farmers’ serum r = 0.364, *p* < 0.01. The mean daily PEF values in farmers were significantly lower compared to controls (469.7 +/− 127.5 vs. 562.9 +/− 123.8; *p* < 0.001; the data were adjusted for gender, height, and smoking). PEF daily variability (amp%mean) was higher in farmers compared to controls (9.3 vs. 8.1%; *p* < 0.05).
Schenker et al. (2005) [24]	Cross-sectional study	89.9% male Median age 54 years Ethnicity: 84.5% white Non-smoker 54%	Logistic regression	Adjusted prevalence odds ratio—persistent wheeze and current smoking status 4.7 (3.1–7.3); persistent wheeze and high dust exposure 1.8 (1.1–3.2); persistent wheeze and live on farm 1.7 (1.1–2.6); persistent wheeze and male sex 2.9 (1.4–6.4); persistent wheeze and asthma per MD 7.7 (5.1–11.8); chronic cough and age (40–59) 2.4 (1.0–5.6); chronic cough and current smoking status 7.3 (4.2–12.5); chronic bronchitis and former smoking status 1.9 (1.0–3.4); chronic bronchitis and current smoking status 5.8 (3.1–10.6); chronic bronchitis and asthma per MD 4.3 (2.4–7.8)
Burch et al. (2009) [23]	Cross-sectional study	100% male Age 25–24 years—46% Race: 70% Caucasian Never tobacco use 51%	Geometric mean and least squares mean	Exposure quartiles 1 vs. 4—dust (mg/m3) and MPO (ng/mL) 57 vs. 21, *p*-value 0.01; endotoxin (EU/mg) vs. IL-8 (pg/mL) 145 vs. 228, *p*-value 0.05; sum of all 3-OH fatty acids (pmol/mg) and MPO (ng/mL) 21 vs. 53, *p*-value 0.01
Adhikari et al. (2011) [22]	Cross-sectional study	57.1% male Mean age 40.1 years Ethnicity: 99% white Current smokers 17.1%	Fisher’s exact test	No significant associations. Usually bringing up phlegm was higher in workers than controls with a crude PR of 4.4, *p*-value 0.133.
Guillam et al. (2013) [29]	Prospective cohort study	60% male Mean age 45.4 Nonsmoker 63.5%	Logistic regression	Respirable dust concentration association with respiratory symptoms: day and/or night cough OR 2.65 (1.16–6.08); chronic cough OR 2.80 (1.12–7.02); chronic phlegm OR 2.07 (1.01–4.27); symptoms of chronic bronchitis OR 4.21 (1.21–14.7).
Madsen et al. (2013) [21]	Cross-sectional study	60.6% male Median age 38.5 years Country of birth—Eastern and Central Europe, Denmark, the Middle East, and Southeast Asia	Pearson’s correlation coefficients (r2)	Exposure to fungi and fungi in NAL r2 = 0.62, *p*-value < 0.0001 Exposure to beta-glucan and glucan in NAL r2 = 0.42, *p*-value < 0.001
Viegas et al. (2013) [19]	Cross-sectional study	60.6% male Mean age 44.5 years Nonsmokers 56.1%	Prevalence	No significant association was found between duration of exposure, and spirometry.
Rodriguez et al. (2014) [30]	Prospective cohort study	43% male Age 41–50 years—33% Country of birth—Mexico 67% Primary school education 58% Current smoker 6%	Multiple linear regression	High TWSRA dust score in past year and FEV6 estimate (SE) 0.22 (0.10), *p*-value 0.04; months worked in agriculture in past year and FEV1 estimate (SE) 0.08 (0.10), *p*-value < 0.001, FEF_25%–5%_ estimate (SE) 0.11 (0.03), *p*-value < 0.001, and FEV6 estimate (SE) 0.11 (0.02), *p*-value < 0.001
Mitchell et al. (2015) [26]	Cross-sectional study	100% male Median age 33.7 years Ethnicity: 90.4% Hispanic Sixth grade or less education 52.8%	Logistic regression (mixed models)	Mixed models for FEV1/FVC and FEF 25–75 adjusted for age and shift time—total endotoxin and FVC, mL 24.46 (−44.65 to −4.27), *p*-value 0.018
Nonnenmann et al. (2017) [25]	Non-randomized cross-sectional study	92% male Mean age 32 Ethnicity: 94% Hispanic Ever smoke 70%	Beta Coefficient (Standard Error)	Relationship between endotoxin and cross-shift pulmonary health measures (FEV1): β (SE) −0.058 (0.039) *p*-value 0.081
Audi et al. (2017) [28]	Prospective cohort study	63.8% male Mean age 47.02 years High school education—58.33% Nonsmokers 87.5%	Mann–Whitney U test	IL cytokine concentration and woken by an attack of shortness of breath 2.3 (*p*-vale 0.009); IL cytokine concentration and COPD 1.1 (*p*-value 0.008)
Sak et al. (2018) [20]	Cross-sectional study	42.9% male Nonsmoking 51.2% 38% no education	Logistic regression	PM2.5 and wheezing OR 2.153 (1.164–3.981) PM2.5 and chest tightness OR 2.211 (1.190–4.108) PM10 and chest tightness OR 1.123 (1.002–1.259)

## Data Availability

No new data were created or analyzed in this study. Data sharing is not applicable to this article.

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
