# Peer review of "A Narrative Review of Occupational Air Pollution and Respiratory Health in Farmworkers"

_ijerph, 2021, doi:10.3390/ijerph18084097_

Round 1
Reviewer 1 Report
Major Comments: This is a nice review of the literature on farm workers and respiratory health. There are however, a couple fixable problems in the paper. First, you use PM2.5 and PM sorta interchangeably throughout the manuscript and in some places where you seem to mean to use the more general 'PM' term. Similarly, you refer to respirable dust or respirable particles incorrectly throughout the paper by using it as a general term for particles that might get inhaled. It actually refers to a specific size range of particles to occupational and environmental researchers (4 µm for occ and something a little smaller for env/EPA). The more important problem with the paper is the proposed focus on infectious disease. While a good focus, none of the 12 reviewed papers discuss infection and the paper gets confusing. I realize that this was there a priori focus/hypothesis, but you jump from respiratory illness outcomes to discussion of no infections and it gets unclear what the paper is about. This should be clarified.
Minor Comments:
- The title is too general and needs to be changed. It says epi studies, but the search was much narrower and didn't include epi studies of agricultural injuries, etc.
- The search period is a little odd. Given the 1998 previous review probably included only 1997 and before, did you miss publications from 1998 and 1999?
- The abstract doesn't mention the main finding of the paper - that no infection studies have been done.
- Should the 2007 pub on infections (ref #10) be updated? Is there nothing after that?
- Redundant: "
This elevated risk suggests this population may be more susceptible to severe acute respiratory infections among crop farmworkers"
- Figure 1 - Useful? Move to Supplement? Also, the minor tick marks at 0.5 intervals are inappropriate for the whole number results.
- This reviewer was perplexed by the use of a parenthesis to define a number and kept thinking is was a ref: "
"One (1) study [29] coupled"
- This should be an area and not distance:
"spread over a 65 km2 distance"
- Confusing:
"The questionnaires derived a time-weighted self-reported average (TWSRA) dust score for ex- posed workers [30] and the other calculated"
- In section 3.3.2, is something missing from this? More lavage fluid doesn't make sense - was more recovered?:
"two- to three times higher mean nasal lavage fluid"
- Is this an error? The Tables says it p = 0.04: "The same study also noted a relationship, though non-significant, be- tween elevated dust score and better lung function (FVC, FEV6)"
- Non-sentence:
"A study conducted among farmworkers who worked in stables and granaries in France [28]."
- Ref #20 had other more significant changed outcomes (in Table). Why was only this one put into the text?:
"associated with higher chest tightness by 1.1-fold"
- Implies that the population found something:
"A similar popu- lation of dairy farmworkers across four states located in Northcentral US, also found"
- Was this supposed to be mg/m^3?:
exposure levels of respirable dust was greater than 0.1 mg [29]
- Table 2: Madsen study - which ref number?
- Table 2: Several numbers in the last column are confusing. Do they denote a direction of change (Rodriguez paper)? Also, many of the results with beta values have only one number in the following parentheses rather than 95% CI.
- Non-sentence:
Although results from the reviewed studies can inform hypotheses and assumptions for prospective (or retrospective) cohort studies
- I have never heard the term 'filtered monitors'.
- 'nano-fibers' seems too specialized and no mention is made of the 'regular' inhaled fibers.
- Add 'is'?
the latter considered
- Add 'more'?
when done correctly self-reporting is useful in giving a range of responses than other data collection instruments
- Redundant:
For example, US-hired crop farmworkers who are exposed to high concentrations of occupational PM2.5, which may include a com- plex mixture of hazardous chemical and biologic components, are also exposed to other environmental and psychosocial stressors. In addition to enhanced susceptibility from ag- ricultural PM2.5 exposure, farmworkers can experience specific vulnerabilities that could exacerbate ARI severity.
24. The comment about spirometry is valid but harsh - proper expertise and calibration take care of such concerns.
25. Again, there's confusion on the paper's focus given there are no papers on infection:
Another limitation is that we did not conduct a meta-analysis which would have quantitatively determine effect size across studies
Reviewer 2 Report
I find the paper did set out to highlight on the need to encourage more research among the target group especially in continent not covered as well as advancing new form of technology in the field of exposure assessment.
However there are few suggestion/observation made to help improve the quality of the work:
i. To help improve the material and method and provide clarity on process done to arrive at 12 papers considered here, it will be helpful if the authors consider the use of flow diagram
i.e. PRISMA to show flow of information considered at each phase of the data mining process leading to the final work retained.
ii. In addition, the use of table to present the exclusion/inclusion criteria will help carry the reader along and increase interest on the paper
iii. The first three sentence in result section fit well with the materials and methods than result. Recommend moving this accordingly
iv. There are few structural issues with the presentation which can easily be amended from another round of proof reading.
I chose minor correction as I thought these are issues that can easily be handled among the team to help improve the work quality further.
Reviewer 3 Report
Thank you for the opportunity to review the presented paper. I food it very informative and elegant in the methodology applied. The twelve studies identified in this narrative systematic review included farmworkers from the crop and livestock sectors. Studies consistently showed that farmworkers occupationally exposed to elevated levels of respirable dust had a higher prevalence of respiratory symptoms and adverse conditions such as chronic bronchitis and asthma, and decreased lung function.
The authors not only clearly summarized available data but also identified several gaps in the collected material. One of the major ones is a finding, although personal air sampling methods were predominantly employed, none of the studies characterized PM chemical composition of the dust.
They also underline that, in addition to respirable PM, farmworkers can be exposed to an array of other air contaminants such as toxic gases, bioaerosols, and endotoxins that act as respiratory irritants.
I fully agree with one of the recommendations made, that prospective cohort studies that examine airway respiratory infections are very much need in the future.
The only suggestion I have is to try to make the whole paper shorter. I leave it to the authors to make a decision on how to do it. One of the solutions would be to divide the paper into two – approaching livestock farm or a crop farm separately.
Minor remark
Figure 4. Respiratory condition assessed – in the graph data is presented, to sum up at 100%. It makes be appropriate when there is some overlap in the scope of the studies. From the description above I found that at least two studies look at COPD and asthma (reference 20 and 28) and examined pneumonia, asthma, and emphysema (22).
